# Current Updates on the Role of Microbiome in Endometriosis: A Narrative Review

**DOI:** 10.3390/microorganisms11020360

**Published:** 2023-01-31

**Authors:** Hooi-Leng Ser, Siu-Jung Au Yong, Mohamad Nasir Shafiee, Norfilza Mohd Mokhtar, Raja Affendi Raja Ali

**Affiliations:** 1Department of Biological Sciences, School of Medical and Life Sciences, Sunway University, Bandar Sunway 47500, Malaysia; 2Department of Obstetrics and Gynaecology, Faculty of Medicine, Universiti Kebangsan Malaysia, Cheras 56000, Malaysia; 3Department of Physiology, Faculty of Medicine, Universiti Kebangsaan Malaysia, Kuala Lumpur 56000, Malaysia; 4School of Medical and Life Sciences, Sunway University, Bandar Sunway 47500, Malaysia; 5Gut Research Group, Faculty of Medicine, Universiti Kebangsan Malaysia, Cheras 56000, Malaysia

**Keywords:** endometriosis, vaginal microbiome, microbiome-based therapeutics, gut microbiome, precision medicine

## Abstract

Endometriosis affects approximately 6 to 10% of reproductive-age women globally. Despite much effort invested, the pathogenesis that promotes the development, as well as the progression of this chronic inflammatory disease, is poorly understood. The imbalance in the microbiome or dysbiosis has been implicated in a variety of human diseases, especially the gut microbiome. In the case of endometriosis, emerging evidence suggests that there may be urogenital-gastrointestinal crosstalk that leads to the development of endometriosis. Researchers may now exploit important information from microbiome studies to design endometriosis treatment strategies and disease biomarkers with the use of advanced molecular technologies and increased computational capacity. Future studies into the functional profile of the microbiome would greatly assist in the development of microbiome-based therapies to alleviate endometriosis symptoms and improve the quality of life of women suffering from endometriosis.

## 1. Introduction

According to a recent study, 196 million women between the ages of 12 and 52 have endometriosis, an uncomfortable condition that is estrogen-dependent and causes chronic pelvic and lower abdominal pain [1]. This gynecological illness, which is characterized by the growth of endometrial glands and stromal cells both inside and outside the pelvic cavity, has a detrimental effect on the quality of life of its patients. It affects approximately 6 to 10% of women of reproductive age globally [2,3,4]. Meanwhile, postmenopausal endometriosis occurs in approximately 2–5% of postmenopausal women and this form of the disease has been proposed to have a more convoluted pathophysiology than the premenopausal form [5,6,7,8]. Nonetheless, the illness frequently manifests as dyspareunia, infertility, dysmenorrhea, and excruciating pelvic discomfort [9]. Endometriosis has been the subject of innumerable scientific and clinical studies, but its origin remains unknown [10,11,12]. The most plausible hypothesis is Sampson’s theory, which states that the focus of the disease is caused by retrograde menstruation [13]. The gynecologist John Albertson Sampson suggested that the retrograde tubal flow plants menstrual endometrial tissues in the peritoneal cavity as well as other organs. In contrast to this finding, many scientists have hypothesized that additional factors, including genetic, anatomical, endocrine, inflammatory, and environmental factors, may affect tissue implantation. This is due to studies showing that while 90% of women had retrograde menstruation, only 10% acquire the condition [14,15,16,17,18,19,20,21]. Additionally, researchers discovered that the microbiome and endometriosis development are bidirectional related, implying that any change in the host’s microbiome can have a significant impact on the development and progression of endometriosis [22]. Therefore, it is of utmost importance to thoroughly investigate the link between the microbiome and endometriosis in order to better understand the condition and maybe develop medications that would alleviate the burden on sufferers.

The quantity of bacteria in the human body was once thought to be ten times greater than that of human cells [23,24]. However, a recent study has proposed revised estimations for the number of human and bacteria cells in a human adult (70 kg body weight) and the ratio of bacteria to human cells is now recorded as 1:1 ratio. These findings then highlight the potential impact bacteria cells can have on human health, on top of their “supporting” role in host metabolism and maturation of the immune system. The human gut microbiome, for instance, produces vitamin K and B12, which helps to maintain the integrity of the intestinal mucosa, repairs epithelial cells, stimulates angiogenesis, and modifies the immune system [25,26]. Furthermore, it has been established that the composition of the microbiome at a given site (i.e., gut, vaginal, nasal, etc.) varies between a healthy individual and a patient with specific diseases. A number of these diseases include autoimmune disease, cancer, metabolic cancer, inflammatory bowel disease, and many others [27,28]. There is growing evidence that endometriosis patients have higher levels of bacterial colonization in their endometrial tissues and menstrual blood than do women in the general population [16,29,30,31]. The use of animal models in endometriosis research is critical for understanding the pathophysiological mechanism underpinning the disease [32]. Recently, endometrial tissues have recently been transplanted to ectopic sites to create endometriosis models in tiny laboratory animals such as rodents. This method offers a more affordable alternative than using non-human primates, but it has certain drawbacks because rodents do not naturally contract the disease while some primates do [33]. Without a doubt, pre-clinical studies employing animal models allow researchers to comprehend the mechanics underlying disease and to test potential therapeutic or preventive agents, especially the degrees of experiment safety before carrying out clinical trials on humans. Thus, the aim of this narrative review is to summarize recent scientific and clinical findings on the relationship between endometriosis and microbiomes of different body sites including the female reproductive tract, gastrointestinal as well as the peritoneal region. Ultimately, these discoveries would provide a foundation for future research to carry on the fight against this agonizing ailment, particularly via biomarker discoveries and microbiome-based therapeutics.

## 2. How Common Is Endometriosis?

Before discussing possible important risk factors that could potentiate the development of endometriosis, it is important to understand the prevalence of the disease across the globe, which further enhances our knowledge of the disease burden among different countries. While it is accepted that endometriosis could affect women of any age, a cross-sectional online survey conducted in Canada estimated the prevalence of endometriosis in Canada to be 7.0% (2004 women of 28,532 women surveyed), with nearly half of the respondents (47.5%) aged 18–29 years old when they were diagnosed with endometriosis [34]. Interestingly, the study also reported that 84.1% of women had endometriosis symptoms prior to diagnosis, indicating a considerable clinical burden and perhaps unmet needs for earlier diagnosis.

Fuldeore and Soliman [35] surveyed women aged 18–49 years old in August–November 2012 and the team reported that the overall prevalence of diagnosed endometriosis was estimated at 6.1% (2922 out of 48,020), with a surprisingly large proportion of these respondents (86.2%) reported to have experience symptoms prior to diagnosis [36]. The primary analysis of the retrospective cohort study revealed a declining incidence rate (which includes participants aged 16 to 60 years during the selected study period, who had a uterus, were continuously enrolled for at least two years before the study was conducted, and had at least one healthcare utilization from 30.2 per 10,000 person-years in 2006 to 17.4 per 10,000 person-years in 2015. The incidence rate did not differ by race or ethnicity groups, but the highest incidence rate was observed among women aged 36 to 45 years old. On the other hand, a study in Spain reported prevalence data collected from the year 2009 to 2018 reflected the opposite trend, with the team observing an increased overall prevalence of endometriosis, reaching as high as 1.24% in 2018. The median incidence rates were reported as 94.9 (range: 92.6–102.9) per 100,000 women-years. As reported by Christ et al. [36], Medina-Perucha et al. reported that the incidence of endometriosis was highest among women aged 35–44 years old [37]. Additionally, Yamamoto et al. were interested to understand if ethnicity/race contributes to the endometriosis prevalence among IVF patients, and they discovered a significantly higher prevalence among Asian women compared to Caucasians (15.7 vs. 5.8%, *p* < 0.01) [38]. Asian women had 2.96 times the odds of being diagnosed with endometriosis (95% CI 1.65, 5.31; *p* = 0.0003), whereas Hispanic women and other races/ethnicities did not differ significantly from Caucasians. A systematic review published by Bougie et al. in 2019 found that Asian women were more likely to have the diagnosis of endometriosis (OR 1.63, 95% CI 1.03–2.58, based on 10 articles which explored the likelihood of endometriosis diagnosis in Asian women) [39].

Spanning over to Africa continent, a 10-year retrospective cohort study (from June 2003 to November 2014) at Nordica Fertility Center Lagos, Nigeria explained that 2.69% of participants (61/2265) were diagnosis with endometriosis via laparoscopy, with the highest proportion (43/61) of endometriosis patient in the age group of 31–40 years old [40]. These findings undoubtedly raised concerns about the assumption that endometriosis is a “rare” disease among indigenous African women. Another prospective cross-sectional study conducted at Kenyatta National Hospital reported that 4.6% of indigenous Africans had histologically proven endometriosis (95% CI 0.5–18.4) [41]. Several research groups emphasized the potential of underdiagnosis that leads to low reporting of endometriosis cases in Africa continent [41,42,43].

How prevalent is endometriosis in the Asia Pacific region? Back in 1976, Miyazawa observed an interesting phenomenon in Hawaii whereby clinicians reported a high incidence of endometriosis among Oriental women. The study by Miyazawa analyzed the data of gynecologic admission and endometriosis diagnosis from three hospitals (two in Hawaii and one in Japan) which led to the conclusion that indeed there is a higher incidence among the Japanese population [44]. Approximately 10% of gynecologic admission was reported to be associated with endometriosis. Almost 30 years later, another study conducted in Japan attempted to gauge the prevalence of endometriosis in the country via the Japan Nurses’ Health Study [45]. Out of 15,019 participants, 6.8% of them (*n* = 1025) had a self-reported history of endometriosis. Subsequent analysis of the data collected from 862 participants showed that the mean age of endometriosis diagnosis was 32 years old with the highest proportion observed among those aged 32 to 44 years (150/330, 45.5%). Conversely, a study by Feng and the team estimated the trends in endometriosis incidence from 1990 to 2019 were estimated using join point regression [46]. When compared to data from North and South America continents, their results reported a significant age-related effect on endometriosis incidence at a relatively young age: increased risk was seen in the age groups ranging from 15 to 19 to 20 to 24 years, with the latter group having the highest relative risk of 2.54 (95% CI: 2.45, 2.64). Nonetheless, the study reported declining age-standardized incidence rates (ASIR) by nearly 30% in 2019 and that the overall ASIR was −1.2% (95% CI: −1.20, −1.10). A recent study from China that recruited patients between September 2021 and February 2022 reported that endometriosis remains to be one of the most prevalent gynecological diseases [47]. This study found a slightly lower prevalence of endometriosis (4.09%) than studies from other countries. However, the study also outlines additional research is needed to substantiate their findings given that the majority of the participants (97.95%) belonged to the Han race and a nationwide study is required to properly comprehend the disease burden in the country. Rowlands and the team investigated the prevalence and incidence of endometriosis among Australian women by studying national hospital data, along with three administrative health databases [48]. The team discovered that one in nine women had clinically confirmed or suspected endometriosis by the age of 44 after retrieving the health records of 13,508 Australian women who were born between the years of 1973 and 1978 over a period of 20 years. The cumulative prevalence of clinically diagnosed endometriosis was 6.0% (95% CI 5.8–6.2%) at age 40–44 years, which was higher than the other studies.

In 1992, a study in Southeast Asia that was completed by Yamamoto et al. in the United States reported findings that were in line with their findings [38,49]. Arumugam and Templeton described that endometriosis was more prominent among infertile women from Kuala Lumpur, Malaysia than in the United Kingdom (51% vs. 22%, *p* < 0.001). According to a study from Thailand, 30.5% (101/331) of the women who underwent surgery for the benign gynecological disease were found to have endometriosis (mean age: 39.4 ± 17.4 years old) [50]. One would generally assume that comparable incidence rates were reported with data from the Asian continent, given the multi-ethnicity background in this region. Yet, the data on the epidemiology of endometriosis within this region is scarce and more studies are still required to clearly understand the disease burden and any potential “modifiable” risk factors (e.g., body mass index, parity, etc.) that could contribute to the development of endometriosis [51]. Nonetheless, there are emerging working groups in several Southeast Asia countries including Thailand and Malaysia, that are actively advocating early diagnosis of endometriosis, along with knowledge dissemination and support systems for endometriosis patients [52,53]. While improving general health and empowering the local community, these efforts collectively would allow researchers to understand the etiology of endometriosis before developing an effective management plan to tackle this multifactorial disease.

## 3. Diagnosis and Management of Endometriosis

Despite the fact that endometriosis can manifest itself in various ways, there are suggested diagnostic procedures that clinicians, including gynecologists and general practitioners, should follow [54,55]. One of the ways is observing the patient’s clinical signs and symptoms. An individual with endometriosis may exhibit one or more symptoms, including persistent pelvic discomfort, infertility, dysmenorrhea, dyspareunia, hematuria, dysuria, and dyschezia [56]. Along with pain in the chest and beneath the shoulder blade, other signs that the disease may be present include catamenial pneumothorax, cyclical cough, and cyclical scar edema [57,58,59,60]. During a pelvic examination, clinicians typically feel for pelvic masses, noduling uterosacral ligaments, retroverted fixtures, uterine or adnexal discomfort, and other probable endometriosis symptoms [61,62]. Even though imaging technology such as magnetic resonance and ultrasound make preoperative disease detection possible, it is crucial to understand that a negative imaging result does not necessarily rule out the presence of the disease, particularly if it is a superficial peritoneal disease [63,64]. Transvaginal ultrasound is preferred to gain a clear view of the endometrial and uterine cavity, as well as to identify ovarian endometriotic cysts without ruling out the possibility of deep-infiltrating endometriosis, peritoneal endometriosis, and adhesions related to endometriosis [61,65,66,67,68,69]. Transvaginal ultrasound’s maximum sensitivity is limited to the diagnosis of endometriomas. Nevertheless, with current technological advancements and sufficient training, it may be possible to raise the detection sensitivity for other phenotypes of the disease [70,71]. The laparoscopic examination was previously thought to be the gold standard for diagnosis, but according to a recent European Society of Human Reproduction and Embryology (ESHRE) recommendation state that it should only be used in cases where imaging fails to clearly demonstrate the presence of pathology in suspected patients [72]. Although histopathological confirmation is excellent, it has limitations in terms of its sensitivity because the definitions of the syndrome have been static for decades, which is mostly true for younger women with the illness [72,73]. In fact, up to 40% of laparoscopies performed for pelvic pain were unable to discover any pathology, necessitating the use of alternative non-invasive methods such as biomarkers by clinicians to aid in the early detection of the disease [74]. In order to maximize the likelihood of a correct diagnosis, a combination of numerous diagnostic techniques is clearly required. It is essential that the medical management team can use these diagnostic tools effectively to suggest the best course of treatment to alleviate the patients’ pain.

Endometriosis management calls for an individualized, multimodal, and interdisciplinary strategy [58]. This may include surgical excision of lesions, non-drug therapies, analgesics, hormonal and non-hormonal therapies, or any of those approaches in combination [56,58,75,76]. Table 1 lists treatments for pain while Table 2 lists treatments for endometriosis-related infertility as per recommendations from international guidelines such as the ESHRE Guideline and National Institute for Health and Care Excellence (NICE) Guideline, along with the national consensus or guidelines such as the Thailand Interest Group for Endometriosis (TIGE), the Obstetrical and Gynecological Society of Malaysia (OGSM), the American Society of Reproductive Medicine (ASRM) [52,56,57,77,78]. Researchers highlighted the lack of reliable information regarding the effects of laparoscopic surgery for the contravention of the recommendations, which has led to many doubts and uncertainties in providing patients with a better quality of life and, most importantly, minimizing pain [79]. Non-drug therapies frequently employ dietary intervention, physical therapy, and psychological intervention. Dietary interventions have been shown to have a positive impact on endometriosis patients’ symptoms. In a study, it was shown that omega-3 polyunsaturated fatty acids (o-PUFAs) were found to lower patients’ pain scores [80], demonstrating that they can both lower inflammation and lessen pain [81,82]. Additionally, cognitive behavior therapy underwent trials to determine its efficacy in creating pain coping mechanisms for a variety of chronic pain illnesses with varied degrees of severity [74,83,84]. Another popular non-drug treatment is pelvic physiotherapy, where physiotherapists accompany patients by supervising rehabilitation activities and offer supplementary treatments such as massages to ease their symptoms. The disadvantage of this approach is proving the efficacy of physiotherapy alone in treating the illness, as the majority of studies have evaluated the therapy in conjunction with psychological and/or medicinal therapy [85]. Patient preference, cost, and side effects are routinely considered in the pharmaceutical management of endometriosis. The initial line of treatment is typically analgesics such as non-steroidal anti-inflammatory drugs and hormonal treatment, such as low-dose combination oral contraceptives such as ethyl estradiol and progestins [86]. Moreover, several clinical studies have demonstrated that gonadotropin-releasing hormones (GnRH) antagonists offer potential therapeutic effects in the management of pain [87,88,89,90]. Lastly, non-hormonal therapies such as anticonvulsants, analgesic tricyclic antidepressants, and selective serotonin uptake inhibitors are occasionally prescribed by clinical practitioners to relieve endometriosis patients’ discomfort. The drawback is that little study has been completed to back it up [91]. This information then indicates that there is still much space for improvement in the management of the disease. More study should be completed to ascertain the efficacy of the current treatments to enhance and develop more effective ones.

## 4. The Intricate Relationship between the Female Reproductive Tract Microbiome and Gut Microbiome in the Development and Progression of Endometriosis

The term “microbiome” refers to a collective group of microorganisms that live in a habitat or specific site. Therefore, it is understandable that the microbiome across the human body may demonstrate differences in the microbial population [92]. The classic example would be the gastrointestinal (GI) system (which harbors the largest microbes in the human body)—pH and oxygen availability changes throughout the GI tract which then poses selective pressure on microbes [93,94]. The gastric microbiome of a healthy human adult would consist of a microbial population that could tolerate the low pH in the environment, which include those belonging to the *Prevotella*, *Streptococcus*, *Veillonella*, *Rothia*, and *Haemophilus* genus [94,95]. In contrast, the colon microbiome of a healthy human adult is predominantly colonized by members of bacteria belonging to the genera of *Lactobacillus*, *Akkermansia*, *Enterobacter*, *Lachnospiraceae*, *Prevotella*, and several more [96]. It is important to note that while microbiome structure may vary between individuals, microbial functions are pretty much conserved which allows researchers to exploit them as disease biomarkers or even therapeutic targets as part of the microbiome-therapeutics strategy. The primary function of the human gastrointestinal system was thought to support the host’s metabolism, including digesting, and absorbing ingested nutrients, and excreting waste products of digestion. However, a growing body of evidence is accentuating the role of the gastrointestinal system as an organ of immunity, particularly in maintaining immune system homeostasis [97]. The gastrointestinal-associated lymphoid tissue acts as the “control center” to manage the immune system in response to massive antigen exposure in the gut and activate adaptive immune responses such as B cell maturation [98].

The role of the gut microbiome in the etiology and pathogenesis of the human disease remains one of the top research areas for the past few decades [99,100]. Nevertheless, emerging evidence highlighted the potential of crosstalk between microbiomes of different sites in several human diseases, including the urogenital and gut microbiome given their anatomical proximity [101,102,103]. Comparable to the continuum observed in the gut microbiome, different parts of the female reproductive tract (FRT) display different distributions of microbes [104,105]. In fact, FRT is categorized into a higher part which consists of the endocervix and uterus proper, and a lower part which comprises the vaginal canal and ectocervix (Figure 1). The lower FRT is dominated by *Lactobacillus* spp. and these microbes protect the host against pathogens by creating a low pH environment and production of bacteriocins as well as hydrogen peroxide. The vaginal community state type (CST) classification system describes a total of five CSTs, whereby CST I, II, III, and V are dominated by *Lactobacillus crispatus*, *Lactobacillus gasseri*, *Lactobacillus iners*, and *Lactobacillus jensenii,* respectively [106,107]. These four CSTs are associated with a healthy vaginal microbiome, whereas CST IV, which presents higher proportions of strictly anaerobic bacteria (e.g., *Prevotella*, *Dialister*, *Atopobium*, *Gardnerella*, *Megasphaera*, *Peptoniphilus*, *Sneathia*, *Eggerthella*, *Aerococcus*, *Finegoldia*, and *Mobiluncus*), is suggested to be linked with inflammation or dysbiosis in the vagina. Additionally, it is important to note that vaginal CSTs can change throughout women’s lifetimes. Given that some microbes can stimulate the immune system to trigger inflammation while a portion of them helps to maintain homeostasis in the host by the production of antimicrobial compounds or even immunomodulatory compounds, researchers are now exploring the possibility of microbiome involvement in the development and progression of endometriosis.

### 4.1. Evidence from Clinical Studies: Are There Any Distinct Microbiome Changes in the Vaginal Microbiome?

The majority of the clinical studies investigating endometriosis and microbiome changes were derived from Asia including Japan, China, Taiwan, Korea, and Turkey (Table 3, Appendix A) [21,108,109,110,111,112,113,114,115,116,117,118]. There were also some studies from the United States, Brazil, Sweden, and Canada [119,120,121,122,123]. Almost all the clinical trials conducted diagnosed endometriosis cases via laparoscopy or histology tests and scored based on the criteria described in the Revised American Society for Reproductive Medicine (r-ASRM) classification of endometriosis. The study by Akiyama et al. in 2019 reflected that even though *Lactobacillus* spp. dominated the cervical microbiome of endometriosis patients, they still presented a higher abundance of *Corynebacterium*, *Enterobactericaea*, *Flavobacterium*, *Pseudomonas*, and *Streptococcus* as compared to control (without endometriosis) [21]. Subsequent quantification of bacteria using real-time PCR confirms the finding from next-generation sequencing, in which *Enterobacteriaceae* and *Streptococcus* abundance were statistically different between endometriosis and non-endometriosis control (*p* > 0.05). Similarly, another team in Taiwan described that not only the cervical microbiome of endometriosis patients was different from healthy women, but there were also some differences between endometriosis patients in Stage I and II as compared to those in Stage III and IV [115]. The team has suggested that potential microbial biomarkers for different stages: (a) Stage I–II: *L. jensenii* or members in *Corynbacteriales*, *Porphyromonadaceae,* and *Ruminococcaceae*, (b) Stage III–IV: *Bifidobacterium breve* and *Streptococcaceae* members (e.g., *Streptococcus agalactiae*).

Given the challenges in obtaining cervical specimens without cervicovaginal contamination and the nature of biomass in the upper FRT, several teams have attempted to study the differences in the microbiome of the lower FRT. For instance, three studies in Brazil and China studied the vaginal swab or fluid obtained from patients and observed a lower abundance of *Lactobacillus* in the endometriosis group as compared to the control [114,118,119]. Besides that, the study by Ata et al. discussed the differences in vaginal samples obtained from Stage III or IV endometriosis patients as compared to healthy women [108]. At the genus level, *Gemella* and *Atopobium* spp. was absent in the vaginal samples obtained from the endometriosis group. A similar approach was taken by Perrotta *et al.*, but the team took a broader approach to look at the vaginal CST rather than looking at just a specific group of microbes [120]. These data then allowed the team to build a random forest-based classification model with machine-learning methods on microbiota composition to predict r-ASRM stages of endometriosis. Analyzing the changes during follicular and menstrual phases yielded highly predictive taxa which can be used to predict either stage I-II or stage III-IV endometriosis—the genus *Anaerococcus* (phylum *Firmicutes*).

### 4.2. Beyond the Female Reproductive Tract: Connections between Gut Microbiome, Peritoneal Microbiome, and Endometriosis

On the contrary, Chen et al. were unable to identify microbial signatures from FRT microbiome for use as a biomarker but a prediction of metagenome functions using bioinformatic tools indicated a higher proportion of microbes involved in general metabolism, lipid metabolism as well as synthesis and degradation of ketone bodies [109]. Furthermore, Huang et al. conducted a study in China, that recruited patients from June 2019–October 2019, and reported that there are no significant differences in cervical microbiome observed between women without endometriosis and endometriosis patients [112]. In spite of this, the team uncovered differences in fecal microbiome composition between women with endometriosis and those without. Analysis of the fecal microbiome showed depletion of ten taxa including *Clostridia Clostridiales*, *Lachnospiraceae Ruminococcus*, *Clostridiales Lachnospiraceae,* and *Ruminococcaceae Ruminococcus*, along with an increased abundance of *Eggerthella lenta* and *Eubacterium dolicum* in endometriosis patients as compared to women without endometriosis. Likewise, another report by Shan et al. increased *Firmicutes*/*Bacteriodetes* ratio in endometriosis group, with enrichment of *Actinobacteria*, *Cynaobacteria*, *Saccharibacteria*, *Fusobacteria,* and *Acidobacteria* (*p* < 0.05) as compared to the control [113]. In essence, these results imply the involvement of the gut microbiome in the development of endometriosis.

Apart from the FRT and gut microbiome, there is another special microbiome that has been investigated to understand the development of the progression of endometriosis—the peritoneal microbiome. Once thought to be “sterile”, the peritoneal microbiome is found to be associated with the etiology of diseases such as end-stage kidney disease and cancer [125,126,127]. As such, it is certainly logical to investigate the changes in the peritoneal microbiome among endometriosis patients, given that the lesion may involve the peritoneum [110,116,128]. In 2022, Yuan et al. reported that there was a total of 276 operational taxonomic units (OTUs) detected in peritoneal fluid collected from endometriosis patients (as compared to 211 OTUs in the control group); out of which, 120 of them were unique to endometriosis group [117]. At the genus level, there was a significantly higher abundance of *Acidovorax* (*p* = 0.01), *Devosia* (*p* = 0.03), *Methylobacterium* (*p* = 0.03), *Phascolarctobacterium* (*p* = 0.03), and *Streptococcoccus* (*p* = 0.04) in endometriosis group than the control group. In order to investigate potential crosstalk between different microbiomes, a research group in Korea decided to study the extracellular vesicles in the peritoneal fluid of endometriosis patients in comparison to those without endometriosis [111]. The team successfully characterized microbes present in the extracellular vesicles and reported that there was a significant decrease in *Actinobacteria* at the phylum level among endometriosis patients.

### 4.3. Establishment of Animal Model for Endometriosis

Over the past few years, researchers have developed animal models for endometriosis to study the changes in the microbiome, specifically to understand how microbes are involved in the pathogenesis of endometriosis as well as the influence on disease progression. Based on the published literature, the majority of studies reporting microbiome changes established the endometriosis model via three main ways: (a) homologous (syngeneic) transplantation, (b) autologous transplantation, or (c) heterologous transplantation of uterine or endometrial tissue [33]. Homologous transplantation requires a donor animal of the same species which typically receives hormonal treatment (e.g., estradiol benzoate) prior to sacrifice. Uterine tissue from the donor will then be injected or sutured into the peritoneal cavity of the recipient animal to the development of endometriosis in humans [129]. On another note, autologous transplantation produces comparable endometriosis phenotypes as homologous transplantation, but this process involves surgical intervention and transplantation within the same animal. In simple terms, the endometrial tissue will be recovered from one of the uterine horns and processed before being injected intraperitoneally into the same animal. In fact, the autologous endometriosis model using rats was initially described in 1985 by Vernon and Wilson, where they compared four types of surgical techniques in producing pathophysiologic features that are consistent with endometriosis [130]. In their study, the team discovered that rats that underwent autologous transplantation had higher peritoneal adhesion, with the implants developed into ellipsoidal cystic structures which consisted of both endometrial glands and stroma. The success in developing the endometriosis model created great opportunities for subsequent studies to investigate the role of the microbiome in endometriosis (Table 4) [131,132,133,134,135,136].

Similar to the findings from clinical studies, animal models of endometriosis reflected potential impact on different microbiomes. Most of the studies which attempted to search for microbial signatures in endometriosis were focusing on a specific locality—the gut microbiome which harbors the most microbes in the human body. Interestingly, an earlier study by Yuan and the team in 2018 reported enrichment of Firmicutes and Actinobacteria phylum in mice with endometriosis induced via intraperitoneal injection of endometrial segments, along with an elevated level of *Bifidobactericeae* and *Alcaligenceae* [131]. Consistent findings were also reported by Ni and the team in 2021 where they discovered an increased abundance of *Clostridium*_*sensu*_stricto_1, *Bifidobacterium,* and *Candidatus*_*Saccharimonas*, on top of elevated *Lactobacillus* abundance in the gut microbiome of endometriosis mice [136]. Their team conducted a linear discriminant analysis that further explains the differences in microbial composition between the endometriosis and control group which arises from 17 genera in the former and 11 genera in the latter group.

Furthermore, the ratio of *Firmicutes*/*Bacteroidetes* has been used as a general indicator of a healthy gut microbiome, given that these two populations constitute a large proportion of microbes in the gut. Results from endometriosis studies involving the use of mice reported heterogeneity in results. While most of the studies reported an increase in *Firmicutes* and a reduction in *Bacteroidetes* abundance, there were two studies reporting the opposite effect, observing a drop in *Firmicutes* abundance and an elevated abundance of *Bacteroidetes* [132,134]. Even so, these findings might be attributed to sampling methods as most of the studies collected fecal pellets, rather than feces from the cecum segment. On top of that, findings presented by Chadchan et al. suggested that the administration of metronidazole in mice with induced endometriosis reduces endometriotic lesion growth, which in turn suggests the possible role of metronidazole-sensitive microbes in the disease progression and warranted further studies to devise better management plan for endometriosis [132].

Additionally, the interest in identifying the key microbiome changes that contribute to the pathogenesis and progression of endometriosis continues to encourage scientists to explore animal models other than mice. In fact, a study in China adopted the autologous transplantation methods used in mice to create endometriotic lesions in six-to-eight weeks-old female Sprague Dawley (SD) rats [135]. The results were comparable to the mice model where they reported an increased Firmicutes/Bacteroidetes ratio, with a decrease in abundance of another bacterial family known as *Ruminococcaceae*. Decreased *Ruminoccoccaceae* abundance has been proposed to have a negative impact, given that these strictly anaerobic bacteria can produce anti-inflammatory short-chain fatty acids such as butyrate and contributes substantially to maintaining the general health of the gut [137,138,139,140]. While many studies have also reported the diminished *Ruminococcaceae* abundance in the gut microbiome in autoimmune diseases such as inflammatory bowel disease, several reports emphasized significant inverse correlations of *Ruminococcaceae* and its member, *Ruminococcus* with the inflammatory cytokine IL-6 [141,142].

Another recent study on non-human primates, olive baboons (*Papio Anubis*) in reproductive age also indicated the involvement of T regulatory cells as they investigated changes in the gastrointestinal and urogenital microbiome [121]. Similar to findings from the rodents’ model of endometriosis, the team reported dynamic changes in the gut microbiome of these olive baboons throughout their 15-months study. For instance, the abundance of genera *Succinivibrio*, *Prevotella*, *Megasphaera*, *Lactobacillus,* and CF231 was decreased in fecal samples at three months post-induction of endometriosis; out of these genera, three of them—*Succinivibrio*, *Prevotella*, and *CF231* abundance increased throughout the disease progression from six to nine months post-induction. Among the literature retrieved from databases reporting endometriosis in animal studies, this is the only study that implies the relationship between gastrointestinal, urogenital, and peritoneal microbiome. In the peritoneal cavity, it was reported that a group of unclassified bacteria dominated the peritoneal microbiome followed by *Proteobacteria*. The team reported that both the peritoneal and vaginal microbiome were diminished upon disease induction and failed to be restored throughout the progression of the disease. An in-depth correlation study on microbial species and peripheral immune cells after induction of endometriosis identified that certain microbial populations in the GI tract displayed a positive correlation with immune cell populations (i.e., natural T regulatory cells and T helper 17 cells) at different study time points, but not the peritoneal and vaginal microbiome. Altogether, the authors concurred that additional analysis needed to be conducted to elucidate the exact mechanisms of these bacterial species in modulating host immune response. These exciting findings further strengthen the rationale to exploit microbiomes as treatment targets, especially their potential in modulating the host immune system and potentially alleviating pelvic inflammation which is commonly seen in endometriosis patients.

## 5. Potential Benefits of Probiotics in the Management of Endometriosis

While pharmacotherapy remains critical in the symptomatic management of endometriosis patients, microbiome-based therapeutics may potentially be used in the nearest future to restore the balance in microbiomes and to alleviate chronic inflammation that is commonly observed among endometriosis patients [118,143,144,145]. For instance, *L. gasseri* OLL2809 was found to be able to inhibit the development of ectopic endometrial cells in the peritoneal cavity via activation of natural killer cells in a rodent model of endometriosis, which most likely occurred via the induction of interleukin-12 production [144]. In the subsequent placebo-controlled study, Itoh et al. also found that taking *L. gasseri* OLL2809 tablets for three months significantly reduces pain intensity on the visual analog scale (VAS) and dysmenorrhea on the verbal rating scales (VRS) in the active group, recording at −3.28 ± 0.36 and −1.44 ± 0.17, respectively, as compared to the −2.00 ± 0.29 and −1.03 ± 0.16 recorded in the placebo group [146]. Correspondingly, a different study conducted in Japan supported the use of *L. gasseri* OLL2809 in the treatment of the condition. In this study, endometriosis volume was significantly different between the active group of rats and the control and dienogest-treated groups, with a significant difference in log value of *p* < 0.05 recorded after four weeks of treatment [147]. Another pilot placebo-controlled randomized clinical trial in Iran evaluated the effects of a multi-strain probiotic capsule known as LactoFem^®^ capsule (containing 10^9^ colony of *Lactobacillus acidophilus*, *Lactobacillus plantarum*, *Lactobacillus fermentum,* and *L. gasseri*) among Stage III-IV endometriosis patients. The team noted a significant drop in dysmenorrhea scores after 8 weeks of treatment in the probiotic group—from 6.53 ± 2.88 to 3.07 ± 2.49 as compared to 5.60 ± 2.06 to 4.47 ± 2.13 (*p* = 0.018). Then again, more studies should be conducted to evaluate the actual changes in different microbiomes post-administration of probiotics to monitor the microbial dynamics throughout the period as well as test different administration routes to ensure optimal results from microbiome-based therapeutics. Given that the methods of probiotics preparation differ between administration routes, considerations should also be given to its stability to ensure optimal delivery to the targeted site [148,149]. Along with that, the actions of probiotics on microbiome stability are equally important to counter dysbiosis and subsequently provide beneficial effects in a long-term manner. To date, there is still a lack of guidelines outlining or supporting the standard use of probiotics in the management of endometriosis. Thus, additional investigations into these aspects would enhance the understanding of disease etiology as well as strengthen the rationale for using microbiome-based therapeutics in endometriosis management.

## 6. Future Recommendations and Conclusions

In the past few decades, the scientific community witnessed the advancements in molecular techniques that allow in-depth investigations on host biology as well as the involvement of microbes in human diseases. As a matter of fact, understanding a complex disease such as endometriosis requires more than just perseverance and collective efforts from experts in different fields. Based on the gathered literature, there were some levels of changes in microbial signatures present among endometriosis patients; interestingly, these changes were not only restricted to the FRT but also at other sites including the gut as well as the peritoneal region. Yet, it is evident that the search for microbial-derived disease biomarkers for endometriosis is still in the infancy stage, especially for its implementation in clinical applications. Further studies would need to be conducted to resolve the heterogeneity issue observed in different clinical studies, given that these differences in data may arise from several factors including ethnicity, diet, or specimen quality [150,151,152]. Functional analysis of the microbiome using whole genome shotgun metagenomic sequencing can possibly provide more insights into the functional characteristics of specific microbes in the specimens.

Another essential point to consider in the crosstalk of microbiome and host processes is the estrogen-microbiome axis [121,124,153,154,155]. While studies have noted the relationship between the microbiome and immune response, emerging data is reflecting that alterations in the estrobolome brought upon by dysbiosis can trigger estrogen-mediated pathologies, including endometriosis as well as endometrial cancer. On top of identifying key microbial signatures that could be used as biomarkers to predict disease development or progression, understanding the functional role of these microbes would indefinitely assist in the design of an effective management plan for endometriosis. All in all, there is still much to be explored but global cooperative efforts between government authorities, researchers, clinicians, and the public will pave the way in tackling a chronic, “neglected” disease such as endometriosis.

## Figures and Tables

**Figure 1 microorganisms-11-00360-f001:**
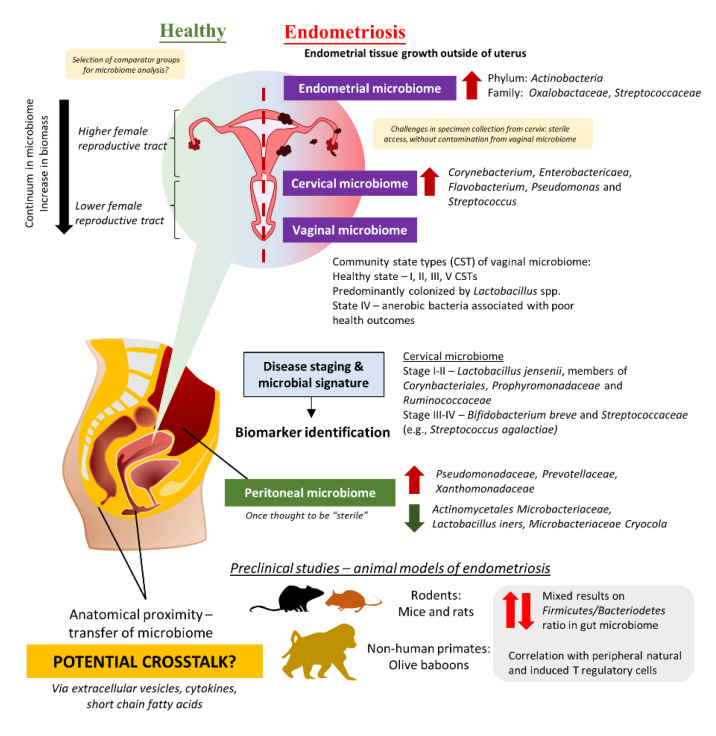
The potential role of the microbiome in endometriosis.

**Table 1 microorganisms-11-00360-t001:** Examples of endometriosis guidelines and their level of evidence in treating pain (GPP: good practice points) [52,56,58,77,78].

Treatments for Pain			Endometriosis Guidelines				
			ESHRE ^a^ (2022)	TIGE (2022)	NICE (2017)	OGSM (2016)	ASRM (2014)
Surgical	Ablation		••	✓	✓	✓	✓
	Excision		••	✓	✓	✓	✓
	Excision of Ovarian Endometrioma		••	✓	✓	✓	✓
	Ablation of Ovarian Endometrioma		•	✓	✓	✓	✓
	Excision of Deep-Infiltrating Endometriosis		••				
	Hysterectomy		••	✓	✓	✓	✓
	Adhesiolysis (Anti-adhesion Agents)		X				
	Presacral Neurectomy		X				(SE)
	Laparoscopic Uterosacral Nerve Ablation						
Pharmacological	Combination Oral Contraceptives Pills		••	✓	✓	✓	✓
	Danazol		X	✓ (SE)		✓ (SE)	✓ (SE)
	Dienogest		••	✓	✓	✓	✓
	Medroxyprogesterone Acetate		••	✓	✓	✓	✓
	Levonorgestrel-Releasing Intrauterine System		•••	✓	✓	✓	✓
	Anti-Progesterone (Gestrinone)		X				✓
	Analgesics (non-steroidal anti-inflammatory drugs)		•	GPP	GPP		
	Aromatose Inhibitors		••				✓
	GnRH-Agonist		••	✓	✓	✓	✓
	GnRH-Antagonist		•••				
	Selective Progesterone Receptor Modulators		•••				
	Selective Estrogen Receptor Modulators		•••				
	First-Line of Treatment		Not Recommended ^d^	••	Low Evidence Level	X	Removed by GDG ^g^
	Second-Line of Treatment		No Recommendation ^c^	•••	Moderate Evidence Level	SE	Critical due to Side-Effect ^f^
	Third-Line of Treatment		Insufficient Evidence	GPP	Good Practice Point ^b^	SC	Special Cases ^e^
	Additional/Other Treatment	•	Very Low Evidence Level	✓	No Information of Evidence	M	Mild Endometriosis

^a^ The development of the ESHRE by the Guideline Development Group (GDG) categorized the quality of evidence using the GRADES approach, which is based on the level of confidence in the true effect compared to the estimated effect. The GRADES approach is categorized into “moderate” (•••), “low” (••), and “very low” (•) confidence level. ^b^ Good practice points (GPP) is given when experts from the GDG support and recommend the therapy. ^c^ The guideline did address the therapy but without stating the level of recommendation. ^d^ The guideline strongly suggests against the usage of the therapy. ^e^ Therapy can be utilized when certain conditions are met. ^f^ Therapy received criticism due to side-effects on the patients. ^g^ Therapy mentioned in the previous version but has since been removed due to new negative findings about the recommendation.

**Table 2 microorganisms-11-00360-t002:** Examples of Endometriosis Guidelines and its level of evidence in treating infertility [56,58,77,78].

Treatments for Infertility				Endometriosis Guidelines			
				ESHRE ^a^ (2022)	NICE (2017)	OGSM (2016)	ASRM (2014)
Surgical	Ablation			•• (M)	✓	✓ (M)	✓ (M)
	Excision			•• (M)	✓	✓ (M)	✓ (M)
	Excision of Ovarian Endometrioma				✓	✓ (M)	✓ (M)
	Ablation of Ovarian Endometrioma				✓	✓ (M)	✓ (M)
	Excision of Deep-Infiltrating Endometriosis						
	Adhesiolysis (Anti-adhesion Agents)				✓		
Pharmacological	Combination Oral Contraceptives Pills			••	✓	✓	✓
	Danazol			X	✓	✓	✓
	Dienogest				✓	✓	✓
	Medroxyprogesterone Acetate				✓	✓	✓
	Anti-Progesterone (Gestrinone)				✓	✓	✓
	Aromatose Inhibitors				✓	✓	
	GnRH-Agonist			••	✓	✓	✓
	GnRH-Antagonist				✓		
	Selective Progesterone Receptor Modulators				✓		
	Selective Estrogen Receptor Modulators				✓		
Assisted Reproduction Techniques (ART)	Intrauterine Insemination (IUI)			•		✓	✓ (M)
	In vitro fertilization/Intracytoplasmic sperm injection			••		✓	✓
	First-Line of Treatment		Not Recommended ^d^	••	Low Evidence Level	M	Mild Endometriosis
	Second-Line of Treatment		No Recommendation ^c^	•••	Moderate Evidence Level		
	Third-Line of Treatment		Insufficient Evidence	GPP	Good Practice Point ^b^		
	Additional/Other Treatment	•	Very Low Evidence Level	✓	No Information of Evidence		

^a^ The development of the ESHRE by the Guideline Development Group (GDG) categorized the quality of evidence using the GRADES approach, which is based on the level of confidence in the true effect compared to the estimated effect. The GRADES approach is categorized into “moderate” (•••), “low” (••), and “very low” (•) confidence level. ^b^ Good practice points (GPP) is given when experts from the GDG support and recommend the therapy. ^c^ The guideline did address the therapy but without stating the level of recommendation. ^d^ The guideline strongly suggests against the usage of the therapy.

**Table 3 microorganisms-11-00360-t003:** Clinical studies reporting changes in the gut, peritoneal and urogenital microbiome (AMEM, adenomyosis-endometriosis; CST, community state type; DIE, Deep Infiltrating Endometriosis; N.A, not available; OCP, oral contraceptives).

References	Year	Study Location	Mean Age (Years Old)	Microbiome Analysis	Findings
Akiyama et al. [21]	2019	Japan	Endometriosis: 33.9 ± 5.7 Control: 32.5 ± 6.0	Cervical Mucus	Gram stain results indicate that the individual variability of bacteria might be present regardless of the presence or absence of endometriosis at any stage of the menstrual cycle. *Lactobacillus* spp. are the most predominant with an increase in population for *Corynebacterium*, *Enterobactericaea*, *Flavobacterium*, *Pseudomonas*, and *Streptococcus* in the endometriosis group. Upon PCR quantification, *Enterobactericaea* and *Streptococcus* are the most significant candidates within the endometriosis group as compared to the control.
Ata et al. [108]	2019	Turkey	^#^ Endometriosis: 28.5 (range: 26–31.3) Control: 27.5 (range: 25.8–30)	Fecal Sample, Vaginal and Endocervical Swabs	The composition within the fecal sample is higher as compared to both the cervical and vaginal samples. At the genus level, *Gemella* and *Atopobium* in the endometriosis group were absent in the vaginal samples. *Atopobium* and *Sneathia* were completely absent in the cervical sample, while *Alloprevotella* was significantly increased in the endometriosis group. Genera *Sneathia*, *Barnesella,* and *Gardnerella* were significantly decreased in the fecal sample of the endometriosis group. *Shigella* and *Escherichia* are more dominant in the stool microbiome of the endometriotic group.
Chen et al. [109]	2020	China	36.07 ± 5.57 (range: 18–45)	Cervical swabs and Vaginal swabs	No significance was found in the alpha diversity of two different locations near the cervix, cervical canal, and posterior fornix. In patients with adenomyosis-endometriosis, *Coriobacteriales* shared the largest proportion, *Coriobacteriaceae* was more dominant than any of the other three groups at the family level and *Atopobium* was greater than any of the other groups in the genus level. Patients with AMEM exhibit a higher abundance of *Atopobium*, *Campylobacter*, *Ezakiella*, *Faecalibaterium,* and *Escherichia/Shigella* as compared to the control group and endometriosis group. At the family level, *Coriobacteriaceae* and *Campylobacteriaceae* are significantly higher in the AMEM group.
Hernandes et al. [119]	2020	Brazil	18–50	Vaginal fluid, eutopic endometrium, and endometriotic lesion	Similar profiles were recorded upon microbiome sequencing of the vaginal fluid, eutopic endometrium, and also endometriotic lesion, with *Lactobacillus*, *Gardnerella*, *Streptococcus,* and *Prevotella* being in abundance. Eutopic endometrium and endometriotic lesion showed lower amounts of detected relative reads compared to vaginal fluids. Vaginal sample shows less diversity as *Lactobacillus* predominates. Microbes in the endometriotic lesion are most diverse and consist of *Lactobacillus, Enterococcus*, *Gardnerella*, *Pseudomonas*, *Alishewanella*, *Ureaplasma,* and *Aerococcus.*
Wei et al. [110]	2020	China	31.47 (range: 23–44)	Vagina swab, posterior vaginal fornix swab, cervical mucus swab, endometrium, and peritoneal fluid	*Lactobacillus* is predominant in the vagina, posterior vaginal fornix, and cervical sample. In the endometrium and the peritoneal fluid, the diversity is much higher with mixtures of *Prevotella*, *Veillonella*, *Atopobium,* and *Veillonellaceae.* Female lower reproductive tract is mainly dominated by *Lactobacillus* with a higher abundance present in the vagina (74.6%%) as compared to the posterior vaginal fornix. *Veillonellacaea* is in abundance in the cervical mucus with *Lactobacillus* noted to be decreasing in abundance. In the endometrium, the mix of *Pseudomonas*, *Acinetobacter, and Vagococcus* made up a significant proportion of the microbiome in endometriosis patients. In the peritoneal fluid, *Comamonas* appeared in both the endometriosis group and also the control group. No distinctive dominant bacteria, indicating a more diverse and complicated microbiome
Perrotta et al. [120]	2020	Brazil	Endometriosis: 34.9 ± 6.8 Control: 35.25 ± 6.9	Rectal and vaginal swab	The distribution of vaginal CST differs between follicular and menstrual phases of the same individual. There was an increase in the number of patients with CST IV microbiomes during menstrual phase for both endometriosis (by 30%) and control subjects (by 25%), while CSTs II and V were lost during the menstrual phase.
Lee et al. [111]	2021	Korea	Endometriosis: 36.20 ± 1.30 Control: 39.40 ± 1.10	Extracellular vesicles in peritoneal fluid	Bray–Curtis beta diversity analysis indicated significant differences in the microbial community in order (*p* = 0.005), family (*p* = 0.003), and genus (*p* < 0.001) between the endometriosis group and the control group. At the phylum level revealed that *Actinobacteria*, *Firmicutes*, *Proteobacteria*, *Verrucomicrobia*, *Bacteroidetes*, *Deferribacteres*, *Fusobacteria*, *Cyanobacteria*, *Tenericutes*, *Armatimonadetes*, *Thermi*, *Euryarchaeota*, *Chloroflexi*, *Spirochaetes*, *Planctomycetes*, *Acidobacteria*, *Gemmatimonadetes*, *Synergistetes*, and *Lentisphaerae* were the most abundant taxa. Significant decrease in *Actinobacteria* at the phylum level in women with endometriosis compared with the controls. Significant decrease in *Actinomycetales* and a significant increase in *Pseudomonadales* at the order level in women with endometriosis.
Le et al. [124]	2021	United States	Endometriosis: 32.5 ± 1.1 Control: 32.6 ± 2.0	Urine, fecal and vaginal swab	Fecal microbiome of endometriosis and control was similar before surgery. Post-surgery, the microbiome community of the GI of endometriosis group who used OCP became more similar to the control group, suggesting the restoration of the gut microbial community. For both gastrointestinal and urogenital microbiome compositions, endometriosis patients receiving OCPs had significantly different bacterial communities than endometriosis not receiving OCPs. During DOS, 17β-estradiol was enhanced in endometriosis patients treated with OCPs.
Svensson et al. [122]	2021	Sweden	Endometriosis: 37.8 (range: 32.8–43.3) Control: 37.0 (range: 32.0–44.0)	Feces (self-collected)	Higher alpha diversity in the control group than endometriosis group (*p* = 4.9 × 10^−5^ 12 genus showed differences in abundance between endometriosis and the control group: *Bacteroidia*, *Clostridia*, *Coriobacteria*, *Bacilli, Gammaproteobacter* Correlation between *Prevotella* and gastrointestinal symptoms including constipation (r = 0.307, *p* = 0.014), bloating and flatulence (r = 0.297, *p* = 0.016) and vomiting and nausea (r = 0.295, *p* = 0.017) Patients with current hormonal treatment reflected a difference in abundance of a genus in the S247 family
Wessels et al. [123]	2021	Canada	Endometriosis: 33.8 ± 5.8 Control: 35.1 ± 3.3	Endometrial biopsy tissue	Significant differences in microbial diversity in endometrial microbiome between endometriosis patients and control, when assessed by the Shannon’s index. Higher proportion of “Others” taxa in Stage 4 endometriosis patients (31.3 ± 3.5) than in control (17.7 ± 2.9) (unadjusted *p* > 0.01) Enrichment of Actinobacteria phylum, *Oxalobactaceae* and *Streptococcaceae* families, and *Tepidimonas* genus in endometriosis patients, while the control group had increased abundance of *Burkholderiaceae* family and *Ralstonia* genus
Huang et al. [112]	2021	China	Endometriosis: 38.3 ± 7.88 Control: 34.0 ± 10.8	Peritoneal fluid, cervical swab, and feces (self-collected)	Different microbial compositions in different body sites of endometriosis compared to control based on PCoA. Fecal samples from endometriosis patients reflected Shannon and Simpson-index estimated microbial richness (*p* = 0.006 and 0.013, respectively). Difference in fecal microbiome composition between early (i.e., Stage I-II) and advanced stage endometriosis patients (i.e., Stage III–IV). No significant depletion or increase in the cervical microbiome between the two groups. Significant differences in OTU of peritoneal fluid from endometriosis patients as compared to control. Increase in abundance in peritoneal fluid of endometriosis patients: *Pseudomonadaceae Pseudomonas*, *Prevotellaceae Prevotella*, and *Xanthomonadaceae Luteimonas* Significant depletion of *Actinomycetales Microbacteriaceae*, *Lactobacillus iners,* and *Microbacteriaceae Cryocola* in peritoneal microbiome of endometriosis patient.
Shan et al. [113]	2021	China	Endometriosis/Control: 32 ± 2, 32 ± 3	Feces (self-collected)	Increased *Firmicutes*/*Bacteroidetes* ratio in the endometriosis group as compared to control. Significant higher abundance of *Actinobacteria*, *Cyanobacteria*, *Saccharibacteria*, *Fusobacteria,* and *Acidobacteria* (*p* <0.05) and decreased abundance of *Tenericutes* in the endometriosis group as compared to the control. 36 genera were unique to the endometriosis group, with the highest abundance genus being *Prevotella_*7 (78.78%). Serum IL-17A was positively correlated with *Bacteroides* abundance (r = 0.89, *p* < 0.05), but negative correlated to *Streptococcus* (r = −0.89, *p* < 0.05) and *Bifidobacterium* (r = −0.89, *p* < 0.05). Serum IL-7 was negatively correlated with *Subdoligranulum* abundance (r = −0.95, *p* < 0.05).
Chao et al. [114]	2021	China	Endometriosis: 39.89 ± 6.24 Control: 38.23 ± 7.80	Vaginal swab	Significant difference in vaginal microbiome diversity between the endometriosis and the control group (*p* = 0.043). An analysis of the similarity test revealed significant difference between endometriosis and the control group. Lower relative abundances of *Lactobacillus* and *Shuttleworthia* in the endometriosis group as compared to the control.
Chang et al. [115]	2022	Taiwan	Endometriosis: 35.4 ± 6.7 Control: N.A.	Cervical swab	Beta-diversity analyses detected differences in cervical microbiome of healthy women and endometriosis patients. Mild differences between patients in different stages of endometriosis (Stage I–II vs. Stage III–IV). Microbial signatures observed in patients in Stage I–II: *Lactobacillus jensenii* or members in *Corynbacteriales*, *Porphyromonadaceae,* and *Ruminococcaceae* Potential biomarkers for patients in Stage III-IV: *Bifidobacterium breve* and *Streptococcaceae* members (e.g., *Streptococcus agalactiae*). Increased abundance of *Prevotella bivia* which was associated with patients in Stage III-IV. Subgroup analysis to identify microbial signature that was associated with DIE showed that patients with DIE displayed higher abundance of *Tenericutes* and *Spirochaetes* among the top 10 phyla, along with increased *Streptococcus* and *Prevotella* of the top-10 genera. Increase abundance of *Caulobacter* sp., *Dialister micraerophilus*, *Fibrobacter intestinalis*, *Treponema berlinense*, *Prevotella intermedia,* and *Helicobacter macacae* in DIE patients compared with those without Patients with high pain scores or higher CA125 levels were found to have increased abundance in phyla of *Actinobacteria*, *Tenericutes,* and *Chlamydiae*
Oishi et al. [116]	2022	Japan	Endometriosis: 37.9 ± 6.4 Control: 35.2 ± 8.6	Vaginal fluid, endometrial fluid, peritoneal fluid, ovarian cystic fluid	Majority of endometriosis patients (*n* = 13) were diagnosed as Stage IV, while the remaining five were diagnosed as Stage III. Similar microbial composition between vagina and endometrium. Significant difference in Shannon index observed between endometriosis and control group for both vaginal and endometrium microbiome Clustering analysis (based on a list of infectious bacteria, *Lactobacillus* and *Bifidobacterium*) showed that majority of profiles where bacteria with the highest abundance in vaginal and endometrial microbiome were others than *Lactobacillus* were derived from the endometriosis group.
Yuan et al. [117]	2022	China	Endometriosis: 35.28 ± 7.24 Control: 33.32 ± 8.04	Peritoneal fluid	A total of 276 OTUs were detected in the peritoneal fluid of endometriosis patients, as compared to 211 OTUs in the control group. 55 OTUs were unique in the control group, while 120 OTUs were unique in endometriosis patients. Significant difference in beta-diversity analysis with unweighted UniFrac distance (*p* = 0.028) between endometriosis and control group At genus level, significant higher abundance of *Acidovorax* (*p* = 0.01), *Devosia* (*p* = 0.03), *Methylobacterium* (*p* = 0.03), *Phascolarctobacterium* (*p* = 0.03), and *Streptococcoccus* (*p* = 0.04) in endometriosis group than the control group. Reduced abundance of *Brevundimonas* (*p =* 0.01) and *Stenotrophomonas* (*p* = 0.04) in endometriosis patients as compared to the control group.
Lu et al. [118]	2022	China	Endometriosis: 36.75 ± 7.11 Control: 35 ± 6.61	Vaginal fluid	Out of 16 patients, 9 patients were classified as advanced stage (i.e., stage 3–4, score 16–40), while the remaining 7 patients were diagnosed as early stage (i.e., Stage 1–2, score 1–15) No significant differences between Shannon index and Chao1 index. Analysis of predominant genera showed an enriched population of *Actinobacteria* genus (especially *Gardnerella* and *Atopobium*), while *Firmicutes* including *Lactobacillus* sp. were reduced in endometriosis patients. Significant difference in vaginal microbiome in endometriosis patients as compared to healthy control based on LDA effect size analysis

^#^ Median age, years old.

**Table 4 microorganisms-11-00360-t004:** Studies reporting microbiome changes in animal models of endometriosis.

References	Year	Animal Used and Age	Endometriosis Model	Specimen and Methods Used to Evaluate Microbiome Changes	Important Findings on Microbiome Changes
Yuan et al. [131]	2018	Female C57BL6 mice	Intraperitoneal injection of endometrial segments	Fecal pellet (7, 14, 28, 42 days post-induction)—16S rRNA analysis (V4 region) on Illumina HiSeq platform	Endometriosis group displayed higher beta diversity index than the control at 42 days after modeling. *Firmicutes* were enriched in endometriosis group, while *Bacteriodetes* were enriched in the control group. At phylum level, the endometriosis group reported to have increased abundance of *Firmicutes* and *Actinobacteria* as compared to the control group. Elevated level of *Bifidobactericeae* and *Alcaligenceae* in endometriosis group Nearly two-fold increase in the *Firmicutes*/*Bacteroidetes* ratio in the endometriosis group as compared to the control
Chadchan et al. [132]	2019	Female C57BL6 mice	Intraperitoneal injection of endometrial segments (autologous)	Fecal pellet (21 days post-induction)—16S rRNA analysis on Illumina MiSeq platform	Groups with administration of antibiotics before/after induction of endometriosis displayed lower lesion mass and volume compared to those who did not receive antibiotics. Fecal sample from endometriosis group without antibiotics had higher abundance of Bacteroidetes and lower abundance of Firmicutes as compared to the control group (i.e., no lesion group) Administration of antibiotics had different bacterial composition—endometriosis group with antibiotics had higher abundance of *Proteobacteria* Disease progression observed in endometriosis which pre-treated with metronidazole group after receiving stool material from endometriosis donor mice
Hantschel et al. [133]	2019	12–16 weeks old female C57BL6 mice (Wild-type and Transgenic TgN (ACTB-EGFP)	Intraperitoneal injection of uterine tissue fragments (biopsy punch)	Fecal pellet (7 and 21 days post-induction)—16S rRNA analysis (V4-V5 region) on Illumina Miseq platform	No significant effect on alpha and beta diversity in the endometriosis group A highly diverse community consists of *Bacteroidales* S24-7group, *Lactobacillus*, *Prevotellaceae* UCG-001 group and *Lachnospiraceae* NK4A136 group with the highest relative abundance No significant changes at genus and family level between sham and endometriosis group No intestinal dysbiosis were recorded in the early phase of lesion formation
Ni et al. [134]	2020	6 weeks old female C57BL6 mice	Intraperitoneal injection of endometrial segments	Feces from cecum segment (21 days post-induction)—16S rRNA analysis (V3-V4 region) on Illumina MiSeq platform	Decrease alpha diversity in gut microbiome in endometriosis group as compared to the control group At phylum level, decrease abundance of *Bacteroides* and *Firmicutes* (*p* < 0.05) and increased abundance of *Proteobacteria* and *Verrumicrobia* (*p*< 0.05) in endometriosis group as compared to control Lower ratio of *Firmicutes*/*Bacteroides* in endometriosis group (2.01) as compared to control (2.25) Analysis of top 20 abundant species at genus level—increased abundances of *Allobaculum*, *Akkermansia*, *Parasutterella* and *Rikenella*; decreased abundances of *Lachnospiraceae*_NK4A136_group, *Lactobacillus* and *Bacteroides*
Cao et al. [135]	2020	6–8 weeks old female SD Rats	Intraperitoneal injection of uterine tissue fragments (autologous)	Fecal pellet (28 days post-induction)—16S rRNA analysis (V3-V4 region) on Ion S5TMXL sequencer	Lower richness and evenness of gut microbiome in endometriosis group as compared to the control Increased abundance of *Firmicutes* and reduced abundance of *Bacteroidetes* and *Proteobacteria* in endometriosis than the control group At class level, higher abundance of Bacilli and lower abundance of *Clostridia* and *Bacteroidia* in endometriosis than the control group At family level, higher abundance of *Lactobacillaceae* and lower abundances of *Ruminococcaceae* and *Peptostreptococcaceae* in the endometriosis group as compared to the control groupIncrease *Lactobacillus* abundance in endometriosis group at genus level when compared to control group
Ni et al. [136]	2021	6 weeks old female C57BL6 mice	Intraperitoneal injection of endometrial segments, fecal microbiota transplant	Fecal pellet (21 days post-induction)—16S rRNA analysis (V4-V5 region) on Illumina MiSeq platform	Increased *Firmicutes* and decreased *Bacteriodota* abundance in endometriosis group as compared to control group at phylum level. Increased *Actinobacteriota* and *Patescibacteria* and decreased *Deferribacterota*, *Campilobacterota,* and *Desulfobacterota* in endometriosis group as compared to control. At genus level, increase abundance of *Lactobacillus*, *Clostridium*_*sensu_stricto_1*, *Bifidobacterium,* and *Candidatus_Saccharimonas* and decrease abundance of *Bacteroides*, *Dubosiella,* and *Muribaculum* in endometriosis group as compared to control Linear discriminant analysis reflected those differences between endometriosis and control groups—arises from 17 genera in the former and 11 genera in the latter group.
Le et al. [121]	2022	Non-human primates (*Papio Anubis*), In reproductive age	Intraperitoneal injection of autologous menstrual tissues for 2 consecutive months during menses	Fecal pellets, urine, vaginal swab, and peritoneal fluid (3, 6, 9, 15 months post-induction)—16S rRNA analysis (V4 region) on Illumina MiSeq platform	Induction of endometriosis caused alterations in mucosal microbiome in both gastrointestinal and urogenital tract Decrease levels of *Succinivibrio*, *Prevotella*, *Megasphaera*, *Lactobacillus,* and CF231 in fecal sample at 3 months post-induction of endometriosis, but increased in *Succinivibrio*, *Prevotella,* and CF231 abundance throughout disease progression from 6–9 months post-induction of endometriosis Analysis of vaginal microbiome showed rich abundance *Porphyromonas*, *Mobiluncus*, *Treponema*, *Campylobacter*, *Prevotella*, and *Streptobacillus* prior to inoculation Induction of endometriosis diminished vaginal microbiome which was never restored throughout disease progression Analysis using Person’s correlation coefficient showed certain microbial species in the gastrointestinal tract and urine to have a correlation with peripheral nTregs and iTregs cells

## Data Availability

Not applicable.

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
