# Peer review of "Current Updates on the Role of Microbiome in Endometriosis: A Narrative Review"

_microorganisms, 2023, doi:10.3390/microorganisms11020360_

Round 1
Reviewer 1 Report
This manuscript addresses the topic of microbiome changes in endometriosis. At first reading, it appears very redundant and confusing. The topic is discussed starting from page 8. Previously, topics already known in the literature are discussed: diagnosis, management of endometriosis, etc. These sections need to be significantly reduced. Furthermore, the introduction should better explain the rationale for this review. Also in the introduction, tha authors talk about endometriosis as a pathology of childbearing age. However, it can also occur in menopause, and this must be emphasized. In this regard, it is advisable to add a recent review: PMID: 34439184. To make the manuscript clearer, I would include subchapters: for example, "gut microbiome changes and endometriosis", "vaginal microbiome changes and endometriosis", and so on. Subsequently, a chapter entitled: "Targeted therapies on the microbiome" should be added where tha authors should talk about pre-probiotics. Finally, the conclusions that should summarize the key points of the review. English must be reviewed by a native speaker.
Author Response
Response to Reviewer 1:
The authors would like to thank the reviewer for the patient and careful evaluation of our work and for providing ideas and corrections that will improve the quality of the manuscript.
Point 1: This manuscript addresses the topic of microbiome changes in endometriosis. At first reading, it appears very redundant and confusing. The topic is discussed starting from page 8. Previously, topics already known in the literature are discussed: diagnosis, management of endometriosis, etc. These sections need to be significantly reduced.
Response 1: Noted with thanks. Redundant sentences have been removed and these sections have been revised accordingly.
Point 2: Furthermore, the introduction should better explain the rationale for this review. Also in the introduction, tha authors talk about endometriosis as a pathology of childbearing age. However, it can also occur in menopause, and this must be emphasized. In this regard, it is advisable to add a recent review: PMID: 34439184.
Response 2: Greatly appreciate the insightful comments by reviewer. The occurrence of endometriosis among postmenopausal women has been added into the introduction section (Page 1, Line 37 – 40).
Point 3: To make the manuscript clearer, I would include subchapters: for example, "gut microbiome changes and endometriosis", "vaginal microbiome changes and endometriosis", and so on. Subsequently, a chapter entitled: "Targeted therapies on the microbiome" should be added where tha authors should talk about pre-probiotics.
Response 3: Authors would like to thank Reviewer 1 for this valuable feedback. Sections and sub-sections have been added accordingly (as follows).
- Sub-section 4.1 – Page 9 Line 321 – Page 10 Line 358
- Sub-section 4.2 – Page 10 Line 359 – Page 17 Line 398
- Section 5 on probiotics – Page 21 Line 489 – Page 22 Line 523)
Point 4: Finally, the conclusions that should summarize the key points of the review. English must be reviewed by a native speaker.
Response 4: Thanks for the kind comments. The conclusion section has been improved accordingly (Page 22 Line 524 – Page 22 Line 550).
The language of the manuscript has been improved accordingly by a native speaker.
Reviewer 2 Report
Manuscript ID: microorganisms-2142240
Type of manuscript: Review
Title: Current updates on the role of microbiome in endometriosis
The aim of the review is to enlighten readers on current scientific and clinical findings on the relationship between the human microbiome and endometriosis.
Comments and Suggestions for Authors:
The manuscript is an interesting review, but requires some considerations.
It should be noted in the title and in the abstract section that this is a narrative review.
When the referenced author is followed by "and team" it should be better unified by "et al".
A review of the acronyms used should be made and their meaning appear in parentheses the first time they appear. Thus, on page 4, line 204 the acronym ESHRE appears for the first time, without including its meaning and yet on page 5, line 220 these acronyms appear again including their meaning (European Society of Human Reproduction and Embryology), which should have been done earlier. Please review all acronyms in the manuscript.
Tables 1 and 2 are difficult to interpret. What is the difference between "Not Recommended ••" and "No Recommendation •••"? What does GDG or GPP mean? It can be understood that the ESHRE Guide (2022) consider IVF/ICSI within the Assisted Reproduction Techniques (ART) as "•• Low Evidence Level"? Please clarify these Tables.
In the Future recommendations and conclusions section it is indicated that "more studies would need to be conducted to evaluate the actual changes in different microbiomes post administration of probiotics", but perhaps there should be more debate on this topic and a more extensive assessment of whether it is appropriate to actually use these treatments freely or should be done in properly planned studies of the that we can obtain more solid results. What do the Clinical Practice Guidelines say about the use of prebiotics or probiotics in this pathology?
Author Response
Response to Reviewer 2
The authors would like to thank the reviewer for the patient and careful evaluation of our work and for providing ideas and corrections that will improve the quality of the manuscript.
Point 1: The manuscript is an interesting review, but requires some considerations.
It should be noted in the title and in the abstract section that this is a narrative review.
Response 1: Noted with thanks. It has been clarified that the current work presents as a narrative review in the title and introduction section.
Point 2: When the referenced author is followed by "and team" it should be better unified by "et al".
Response 2: Noted with thanks. Changes have been applied accordingly throughout the manuscript.
Point 3: A review of the acronyms used should be made and their meaning appear in parentheses the first time they appear. Thus, on page 4, line 204 the acronym ESHRE appears for the first time, without including its meaning and yet on page 5, line 220 these acronyms appear again including their meaning (European Society of Human Reproduction and Embryology), which should have been done earlier. Please review all acronyms in the manuscript.
Response 3: Apologies for the typos and inconsistencies in acronyms. These typos have been corrected throughout the manuscript.
Point 4: Tables 1 and 2 are difficult to interpret. What is the difference between "Not Recommended ••" and "No Recommendation •••"? What does GDG or GPP mean? It can be understood that the ESHRE Guide (2022) consider IVF/ICSI within the Assisted Reproduction Techniques (ART) as "•• Low Evidence Level"? Please clarify these Tables.
Response 4: Authors would like to thank Reviewer for the kind comments. Footnotes have been added to Table 1 and Table 2 for better clarity.
Point 5: In the Future recommendations and conclusions section it is indicated that "more studies would need to be conducted to evaluate the actual changes in different microbiomes post administration of probiotics", but perhaps there should be more debate on this topic and a more extensive assessment of whether it is appropriate to actually use these treatments freely or should be done in properly planned studies of the that we can obtain more solid results. What do the Clinical Practice Guidelines say about the use of prebiotics or probiotics in this pathology?
Response 5: Section 5 have been added to discuss the potential benefits of probiotics in the management of endometriosis (Page 21 Line 489 – Page 22 Line 523). At the time of writing, there is still no guidelines clearly indicating/supporting the standard use of probiotics in the management of endometriosis and further investigations are essential to confirm the long-term beneficial effects of probiotics for this pathology (Page 22 Line 515 – 523).
Reviewer 3 Report
Very nice text. Please check the grammatical errors and typos before publication.
Author Response
Response to Reviewer 3
The authors would like to thank the reviewer for the patient and careful evaluation of our work and for providing ideas and corrections that will improve the quality of the manuscript.
Round 2
Reviewer 1 Report
The authors modified the manuscript on the basis of the criticisms raised. It's now improved and clearer for readers.
Reviewer 2 Report
Manuscript ID: microorganisms-2142240
Type of manuscript: Review
Title: Current updates on the role of microbiome in endometriosis: a narrative review
The authors have made comments on the considerations raised. The study has improved with the changes made by the authors.
An effort could be made to clarify the interpretation of Tables 1 and 2. What does SC, SD, M mean? What is the difference between the colors used? Please, correct them by putting yourself in the reader's situation, to favor their easier interpretation.
References should be thoroughly revised to conform to uniform and appropriate standards for the journal Microorganisms.